

**WOCE-Argo Global Hydrographic Climatology**

Viktor Gouretski

Integrated Climate Data Center,
Center for Earth System Research and Sustainablity,
The University of Hamburg, Grindelberg 5, 20144 Hamburg, Germany

*Correspondence to:* Viktor Gouretski (viktor.gouretski@uni-hamburg.de)

**Abstract.** The paper describes the new gridded WOCE-Argo Global Hydrographic Climatology (WAGHC) (Gouretski, 2018). The climatology has a one-fourth degree spatial resolution resolving the annual cycle of temperature and salinity on a monthly basis. Two versions of the climatology were produced differing by the spatial interpolation performed on isobaric or isopycnal surfaces respectively. The WAGHC climatology is based on the quality controlled temperature and salinity profiles obtained before January 2016 with the
average climatological year being in the range 2008 to 2012.

To avoid biases due to the significant step-like decrease of the data below 2 km the profile extrapolation procedure is implemented. We compare the WAGHC climatology to the one-fourth degree resolution isobarically averaged climatology WOA13, produced by the NOAA Ocean Climate Laboratory (Boyer et al., 2013) and diagnose a generally good agreement between these two gridded products. The differences
between the two climatologies are attributed basically to the interpolation method and to the considerably extended data basis. Specifically, the WAGHC climatology improved the representation of the thermohaline structure both in the data poor polar regions and in several data abundant regions like the Baltic sea, Caspian sea, Gulf of California, Caribbean Sea, and the Weddell Sea. Further, the dependence of the ocean heat content anomaly (OHCA) time series on the baseline climatology was tested. Since the 1950s, the both
baseline climatologies produce almost identical OHCA time series.

**1 Introduction**

The description of the mean state of the Global Ocean has a long history. Since the late 19[th] century the continuously growing net of hydrographic observations resulted in the production of increasingly detailed maps of temperature, salinity and other parameters. All these maps were hand-drawn, often having an
imprint of strong subjective data interpretation. The introduction of computers permitted the accumulation and analysis of large amounts of data and led to the construction of the objectively analyzed maps. The first climatology of the World Ocean by S. Levitus (1982) has become a standard for the oceanographic community. Since then the NOAA Ocean Climate Laboratory has been regularly producing improved versions of the global climatology (Levitus et al., 1994; Levitus et al 1998; Locarnini et al 2010). The last
update (Boyer et al 2013) was based on the hydrographic data over the entire time period from the beginning of the hydrographic observations to 2013.

All NODC Climatologies possess a high degree of consistency using similar quality control procedures and the objective mapping method (Barnes, 1964). The interpolation is performed on a set of standard depth levels, with the response function defining the smoothing inherent in the objective analysis method.
However, as noted by Lozier et al (1994), averaging (smoothing) of oceanographic properties on isobaric surfaces results in the production of water masses with temperature-salinity characteristics different from those of the observed data due to the non-linearity of the equation of state for seawater. In order to avoid this artifact, it has been suggested to average the data on isopycnal surfaces. The objective analysis on density

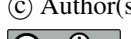


surfaces mimics the process of isopycnal mixing and does not produce artificial water masses. Gouretski and Koltermann (2004) prepared the isopycnally-averaged global hydrographic climatology (WGHC) based on[1] the high-quality data obtained during the World Ocean Circulation Experiment. To achieve reasonable data coverage between the WOCE section lines, selected pre-WOCE hydrographic data were added to the WOCE dataset, which served as a reference dataset for the calculation of the systematic inter-cruise property offsets

(Gouretski and Jancke, 2001). The WGHC was used in a number of applications (for instance, the WOCE hydrographic Atlas of the Atlantic Ocean (Koltermann et al., 2011), the calculation of the absolute salinity (IOC, SCOR and IAPSO, 2010). One of the faults of the WGHC climatology is the absence of seasonality: at all levels only data mean parameter distributions are available. More recently a global monthly isopycnal upper-ocean climatology with an emphasis on preserving a surface mixed layer was created (Schmidtko et

al., 2013).

The purpose of the current study is to produce an update of the WGHC climatology. We use the advantage of the significantly improved data basis due to the implementation of the Argo programme in order to achieve monthly temporal resolution and to increase the nominal spatial resolution to 0.25x0.25 latitude/longitide degrees. We refer to this new climatology as the WOCE-Argo Global Hydrographic Climatology

(WAGHC).

## 2 Constructing the climatology: an overview

Constructing the climatology consists of several steps which are briefly outlined here. First, the climatology time frame, spatial and temporal resolution, and observation types are selected. The automated quality control procedure is applied to the original temperature and salinity profiles, which are subsequently

interpolated on a pre-defined set of depth levels. The interpolated profiles are then averaged in quarter-degree bins on a monthly basis, with the binned data providing the input for the spatial optimal interpolation. Highly smooth gridded fields of water density, temperature and salinity obtained by distance weighted averaging are generated and used as the first guess fields required by the objective mapping method. At each grid-point, the covariance matrices for optimal parameter estimation take account of both

the distance between the data points and the difference in the bottom depth, so that along isobath observations become greater weights compared to across isobath observations. It is assumed that the fields to be analyzed and the noise in the data are uncorrelated. Two versions of the climatology are subsequently constructed in two steps: 1) the isobaric climatology with the optimal interpolation (mapping) performed on depth levels and 2) the isopycnal climatology where mapping is done on local density surfaces. The new

climatology is compared with the last version of the NOAA WOA13 Atlas, which extends over the upper 1500 meters and has the same temporal and spatial resolution. Finally, the new climatology is used as the reference for the calculation of the ocean heat content anomaly time series.

## 3 Data basis

The WOD13 database (including the update as of January 2017) served as the main data source for the

WAGHC climatology. The profiles of the four instrumentation types were used: Ocean Station data (OSD), Conductivity-Temperature-Depth (CTD), Argo profiling floats (PFL), and the Autonomous Pinniped Bathythermograph data (APB). The latter were used only in the Southern hemisphere where data coverage is generally poorer compared to the Northern Hemisphere. All four data types normally report both temperature and salinity. As both of these parameters are required for the spatial interpolation on isopycnal

surfaces, the expendable (XBT) and mechanical (MBT) bathythermograph data were not used. To the total

---



of 4,665,330 temperature/salinity profiles from the WOD13, we have added by 50,848 profiles obtained from the Alfred Wegener Institute, Bremerhaven and 5,340 profiles received from different institutions in Canada. These two data sets helped to improve significantly the data basis for northern polar regions. Table 1 gives details on the data types and data sources contributing to the WAGHC.

Figure 1 shows the yearly number of profiles of each data type retained after quality control. Before 1990 OSD profiles prevail, with CTDs being the main data type between 1990 and 2003. In later years, observations were mostly delivered by Argo floats, with the implementation of Argo floats being marked as a step-like increase in the number of available data.

In general, for most of the Global Ocean, we used data since 1985 (the beginning of the pre-WOCE
hydrographic programme), thus incorporating data within the last 32 years, which is close to the 30-years period for calculating climate norms as recommended by the World Meteorological Organization. In some regions (mostly in high latitudes and several marginal seas) there are still no or not enough modern data, so that older data (since 1925) were taken. However, the time frame for the data selection was narrower, especially within the upper 2 km Argo float depth range.


## 4 Data quality control procedure

Quality control is important for construction of the climatology. Due to the large volume of data, an automated quality control (AQC) procedure was developed. It consists of a suite of quality checks:

1) crude parameter range check

2) spike check

3) constant value check

4) multiple extrema check

5) vertical gradient range check

6) local climatological range check

7) sample depth vs local digital bathymetry check

8) percentage of rejected (flagged) observed levels.

Before quality control, the observed level depths were checked and reordered in increasing order if necessary. For the purpose of initial tuning of the AQC procedure and for the final assessment of data quality, a diagnostic tool was developed which provides statistics of rejected (flagged) data versus time,
observation depth, and bottom depth. The AQC procedure is applied to original profile data separately for temperature and salinity. Table 2 contains statistics of the data rejection rates. According to the statistics, Argo float data are characterized by the lowest rejection rate, followed by CTD, OSD, and APB data. The application of the two quality checks results in the largest percentages of outliers: these are the local climatological range check and the sample level depth vs local bathymetry.

The overall performance of the AQC procedure is illustrated by two-dimensional histograms (Fig. 2). Both for temperature and salinity, the time-depth histograms indicate the decrease of data rejection rates with time. The highest rejection rate is observed around the Second World War and may be attributed to the conditions generally unfavorable for conductiong high quality observations. A significant improvement in





the quality of temperature and especially salinity observations took place with the introduction of CTDs and
electronic salinometers in the beginning of 1970s, with the next data quality improvement due to the
introduction of profiling floats in the mid-2000s. The AQC procedure identified 3.745% and 5.255%
observed levels for temperature and salinity respectively as outliers, whereas for 14.201% and 18.167% of
temperature and salinity profiles, respectively, at least one observed level outlier was identified. Further
details of the quality control procedure are given in the Appendix.

## 5 Vertical interpolation and extrapolation of the temperature and salinity profiles

The quality-controlled observed temperature and salinity profiles were finally interpolated on 65 unevenly
spaced "standard" levels between the surface and 6750m. The depth interval between the levels increases
linearly with depth, so that a better vertical resolution is achieved in the upper layers, where typically higher
vertical property gradients occur. Only levels with both temperature and salinity that pass all quality checks
were retained for vertical interpolation using the weighted-parabola method by Reiniger and Ross (1968).
The interpolation was not performed where the spacing between two levels exceeds the depth-dependent
threshold value.

After the mid-2000s, the majority of the temperature/salinity profiles comes from Argo floats (Fig.1). Since
the floats measure only within the upper 2000 m layer, a step-like decrease in the data coverage occurs
around the 2000 m level, which would create a strong bias towards the observations above 2000m when the
spatial interpolation is performed on isopycnal surfaces. To avoid this artifact, the profile extension method
was developed. The method is based on the observational fact that the local temperature and salinity values
are fairly constant below the main thermocline. First, in the vicinity of each profile potentially suitable for
extrapolation, up to ten deep CTD and OSD profiles are selected and the average deep profile is calculated
using distance weighted mean values at each standard level (the influence radius for the deep profile
selection does not exceed 333km). For the last observed level $Z_M$ (the merging depth) of the profiles subject
to extrapolation the temperature and salinity offsets ($DT_o$ and $DS_o$) relative to the mean profile are calculated.
If the parameter offset for the merging depth does not exceed a predefined threshold value, and if $Z_m$
≥1898m (the deepest WAGHC standard depth level within the Argo depth range), the profile is considered to
be suitable for extrapolation. The average profile is then modified as follows: at each deep level $Z \geq Z_m$ the
offset value $DP=DP_o[1 - (Z - Z_m)/(Z_{max} - Z_m)]$ is subtracted from the average parameter value (temperature or
salinity). The modified mean profile is used to extrapolate the original profile below the level $Z_m$.

Figure 3 illustrates the extrapolation procedure for three arbitrarily selected full-depth CTD profiles. In
order to estimate the average extrapolation error, we selected 52,672 full-depth CTD and OSD profiles
deeper than 2200 m obtained after 1984 and interpolated on standard levels. The extrapolation procedure was
then applied to all these profiles truncated at levels equal or deeper than 1898 m. The respective mean
absolute difference between the extrapolated and the original full-depth profile decrease with depth and are
in the range between 0.03 ℃ at 3000 m and 0.002 ℃ at 6000 m for temperature and between 0.003psu and
0.001psu for salinity ( Fig.4).

Finally, the extrapolation procedure was applied to 720,839 quality controlled OSD, CTD and PFL profiles
obtained after 1984 and having the last level at or deeper then 1898m. For most of the ocean area, the
percentage of extrapolated levels exceeds 20% from the total number of levels, with Argo extrapolated
profiles comprising the largest group. The spatial distribution of full-depth and extrapolated profiles is
shown in Fig.5 along with the percentage of the interpolated levels and extrapolated profile frequency
distributions versus the number of extrapolated levels and the year of observation.





## 6 Temporal and spatial data binning

The significantly increased data basis since the introduction of the Argo floats permits a better time and spatial resolution compared to the earlier WOCE Global Hydrographic Climatology (WGHC) (Gouretski and Koltermann, 2004). For each standard depth surface, the quality controlled vertically interpolated/extrapolated data were gridded by bin-averaging the data separately for each calender month in each one-fourth-degree grid cell. The binning procedure serves two purposes. First, the binning reduces the overall number of observations and, secondly, it reduces noise in the data. The thinning of the input profiles permits the application of the classical optimal interpolation method without the use of a fast multiscale optimal interpolation algorithm proposed by Menemelis et al. (1997).

With the aim to produce a climatology for the most data abundant recent years, the data selection for each spatial bin was performed iteratively, with the data being selected first for the time period 1985-2016, thus imbedding the WOCE hydrographic survey. If no data were available for a particular grid node, the time period was extended to 1957-2016. For a small fraction of the grid-nodes, all data since 1925 were used to produce the bin-averaged values. The percentage of spatial monthly bins populated with two or more observations decreases from 30% in the upper several hundred meters to about 20 % at levels deeper than 2000 m.

The mean climatological year changes with depth, in the range 2007 to 2011. For the upper 2-km layer where the Argo float data prevail, the climatological year is within the 3 year range between 2009 and 2011 (Fig.6). Below the Argo depth range, the mean year is within the range 2007-2009. Differences between the mean climatological year for different calendar months do not exceed one year.

## 7 Spatial interpolation

The contemporary data base does not provide enough data to obtain bin-averaged values for each one-fourth-degree monthly bin, so that an interpolation procedure to fill gaps is needed. The bin-averaged temperature and salinity profiles serve as inputs for the spatial optimal interpolation method, which is used here in the form suggested by Gandin (1964). For the optimal interpolation on isobaric surfaces, the normalized spatial covarience $C_{xyH}$ of temperature and salinity was represented through the negative-squared exponential:

$$C_{xyh} = \exp\text{-}[(r_x/L_x)^2+(r_y/L_y)^2+(h/H)^2], \qquad (1)$$

where $r_x$ and $r_y$ are zonal and meridional distances between the two points, $L_x$ and $L_y$ are the zonal and meridional decorrelation scales, h is the depth difference between the two points and H is the decorrelation depth scale. Outside the +/-20° zonal belt around the equator, $L_x = L_y$ whereas, within this belt, $L_x$ increases linearly from $L_y$ at 20°N and 20°S to $4L_y$ at the equator, in order to account for the zonal elongation of correlation scale within the equatorial belt. The introduction of the $(h/H)^2$ term in (1) represents the added distance penalty for crossing isobaths. The value of H was set to 2 km. Based on the evaluation of the test calculations the noise-to-signal variance ratio was chosen to be 2.0 as the trade off between the smoothness and desirable feature resolution.

For the optimal interpolation on isopycnal surfaces the normalized spatial covarience $C_{xyz}$ was also represented through the negative-sqared exponential:





$$C_{xyz} = \exp\!-[(r_x/L_x)^2+(r_y/L_y)^2+(\mathbb{R}/Z)^2], \qquad\qquad (2)$$

where $\mathbb{R}$ is the depth difference between the vertical positions of the same isopycnal surface, with Z=250m
being the decorrelation depth scale. Introduction of this term is aimed to reduce the depth bias appearing near
the boundaries of the domain, where observations are biased to one side (above or below) of the analyzed
grid level. The objective analysis is performed on deviations between the observations and first guess
values. To provide the distance-weighted means for the first-guess temperature, salinity and density fields,
formula (1) was used with $L_x$ and $L_y$ set to 555 km.

The spatial covariances of the analyzed temperature and salinity fields should be derived from available
observations. However, the correlation length scale must be at least larger than the data spacing (Nuss and
Titley (1994); Sokolov and Rintoul (1999)). We use the mean average distance to the four nearest bin-
averaged neighbor profiles as the measure of data sparseness (Fig. 7). The distance between the observation
points increases with depth from about 70-100 km within the upper 2000 m to about 200-300 km in the
lower layer. These mean values were used as a guide for the choice of decorrelation length scale for the
optimum interpolation. After some experimenting, we decided on a decorrelation scale value of 333 km,
which was used at all levels for the current version of the climatology (the decorrelation scale is the distance
at which the autocorrelation function decreases to 1/e times the value of the zero lag). As noted by Sokolov
and Rintoul (1999) the optimal interpolation produces a spatial average of the analyzed parameters, acting as
a low-pass filter.

During the first step, the isobaric climatology is constructed with the binned data being spatially interpolated
at preselected standard levels for each calendar month. We do not perform spatial interpolation for
temperature and salinity separately. Instead, to avoid the undesirable effect of artificial water mass
production, we first perform spatial interpolation of sea water density. Subsequently the optimal estimate of
temperature on *isobaric surfaces* is obtained. The interpolation of salinity is not performed: the salinity is
inferred from the isobarically interpolated density and temperature values. We note, that the approach
described above differs from the method used for the construction of earlier versions of the World Ocean
Atlas (Levitus and Boyer, 1994; Levitus et al., 1998)), where isobaric interpolation (averaging) was
performed separately for temperature and salinity. The calculated density profiles are checked for hydrostatic
stability and the stabilization is performed if necessary by introducing small adjustments to temperature and
salinity.

At each grid location, the stabilized isobarically averaged density profile defines the set of local density
surfaces on which interpolation of temperature is subsequently performed. As in the isobaric case, the
salinity is inferred from density and temperature values.

The advantage of the isobaric method is that it can be applied in exactly the same way throughout the water
column. However, the averaging (smoothing) of data along levels of constant depth does not correspond to
the process of the water mass mixing in the real ocean which takes place along isopycnal, or more correctly
along the neutral density surfaces. In contrary, the isopycnal averaging does not produce artificial water
masses, but in the regions where isopycnals outcrop at the surface or bottom, the isopycnally averaged
parameters are biased toward the ocean interior (Schmidtko et al., 2013).

## 8 Isobarically- versus isopycnally-averaged WAGHC climatology

Differences between parameter distributions on selected levels between the isopycnally- and isobarically-
averaged WAGHC climatologies are shown in the Fig. 8 a-e. As expected the largest differences occur in



regions of strong spatial temperature and salinity gradients, like the Gulf Stream, Kuroshio, Antarctic Circumpolar Current, equatorial and tropical Pacific Ocean. In such regions the absolute difference in temperature and salinity can exceed 1°C  and 0.2PSU respectively. The differences diminish with  increasing depth. Thus, at the level of 1898 m, only the North Atlantic and the belt of the Antarctic Circumpolar Current show systematic differences exceeding 0.05°C and 0.01 psu for temperature and salinity,

respectively.  Integrated over the whole ocean area (Fig. 8g),  the climatological isobaric temperature values are higher than the isopycnally averaged values. The same is true for the salinity except for the upper 100-meter layer. Below 2000 m, typical absolute differences between the isobarically and isopycnally averaged temperature and salinity values remain below 0.25°C and 0.005PSU respectively.

## 9. WAGHC versus WOA13 climatology

We compared the WAGHC monthly temperature and salinity fields with  respective fields from the NOAA

WOA13 atlas (Boyer et al., 2013).  This atlas represents the last version of the NOAA temperature and salinity climatologies. For the upper 1500 meters, the 1/4-degree resolution WOA13 climatology was used, below only 1-degree resolution WOA13 climatology is available.

### 9.1 Temperature and salinity distributions at levels

As already noted the interpolation in WOA13 is performed on isobaric surfaces separately for temperature and salinity, so that similar difference patters were identified as in the case of isobarically and isopycnally averaged WAGHC climatologies. Indeed, Fig. 8 a-f  and Fig. 9 a-f reveal several qualitatively similar patterns, indicating the largest differences in the areas with strong spatial gradients. We note that part of the differences should be attributed to climate change, since  both climatologies are on average about 27 years

apart. As  progressive warming was observed for the Global Ocean during the last decades Fig. 9 a-f in contrast to  Fig.8 a-f are dominated by the regions of positive temperature differences. These differences are described further below in a separate section.

Shown in Fig.10 a,b are the temperature and salinity differences between the isopycnal and isobaric versions of the WAGHC climatology at 150 m level for  part of the North-West Atlantic Ocean. Here, along the path

of the Gulf Stream, very high lateral temperature and salinity gradients occur with the effect of the data averaging method being especially pronounced. Parameter differences between the WAGHC and the WOA13 climatologies for the same level are presented in Fig 10 c, d. A very good agreement between the respective difference fields is clearly seen, suggesting that the differences between the WAGHC and WOA132 climatologies are mostly due to the difference in the interpolation method.


### 9.2 Differences in temperature-salinity space

The differences between the WAGHC and the WOA13 atlas can be further identified using the classical scatter T, S-diagrams. Shown in Fig. 11 are scatter T, S-diagrams based on the gridded data for six selected depth layers. The largest differences between the two climatologies are found within the upper 1500 m layer,

with the WOA13 showing usually broader T, S-sequences compared to the WAGHC climatology. Unrealistically high WOA13 salinities exceeding 35.5 PSU are found in the temperature range below 2°C.  A generally good agreement is observed for the levels below 1500m where the WOA13 climatology is available on 1x1-degree grid.



To permit a more detailed comparison of the thermohaline properties for both climatologies, we selected 34
regions within the World Ocean (Fig. 12) . Below we describe the most pronounced differences revealed by
scatter diagrams for particular areas (Fig. 13 a,b). Unfortunately, it is not possible to give a definite
explanation for these differences, as many details regarding the construction of the WOA13 are not known to
us. For the Arctic Ocean without the marginal seas, the WOA13 climatology produces unrealistically high
salinities exceeding 36PSU. In contrast, for the Baffin Bay, Kara Sea, White Sea and European Nordic seas,
WOA13 gives much lower salinities compared to the WAGHC climatology. For temperatures below ca. 2$^\circ$C,
the WOA13 climatology gives unrealistically high salinities for the Kara Sea, White Sea, and the Hudson
Bay. At least part of the differences described above can be attributed to the much poorer WOA13 data basis
for the North Polar region compared to the WAGHC.

Significantly different T,S-diagrams are found also for several of the data abundant regions. For instance the
Baltic sea is characterized by extraordinary good data coverage. However, significant deviations between
the two climatologies are clearly seen: 1) the waters with salinities below 5PSU are completely absent in the
WOA13; 2) salinities higher than 25 PSU are not known for the Baltic sea but are present in the WOA13
climatology. Very different T,S-diagrams are found for the Caspian Sea. Here, the WOA13 gridded fields
report salinities lower than 7PSU throughout the whole temperature range, along with unrealistically high
temperatures exceeding 30$^\circ$C. For the Mediterranean Sea the WOA13 gridded product exhibits low-salinity
sequences (below 36.5PSU) not supported by the observational data. In the Pacific Ocean we note the
unrealistically broad WAGHC salinity range for the Sea of Okhotsk, especially for the deep waters with
temperatures below 5$^\circ$C. The T, S-diagrams for the two climatologies differ considerably for the Gulf of
California. Here the WOA13 climatology exhibits a very broad salinity range even for the deep part of the
water column, with temperatures below 12$^\circ$C where the T, S-relation becomes very tight. Similar to the Gulf
of California, we find WOA13 salinity ranges that are too broad in the deep waters of the Gulf of Mexico
and of the Caribbean Sea. The WOA13 climatology is also biased to low upper layer salinities in the
Andaman and Java seas. Finally, we note a broader WOA13 salinity range for the Weddell Sea. Here, the
WOA13 climatology gives unrealistically high salinities exceeding 35PSU, in disagreement with
observations.

### 9.3 Volume-averaged temperature and salinity differences

As noted above, the spatial patterns of temperature and salinity differences at selected levels between the
isopycnally averaged WAGHC climatology and isobarically averaged WOA13 climatology resemble the
differences between the isopycnally and isobarically averaged WAGHC climatologies, suggesting the
dependence on interpolation method.

Shown in Fig. 14 are the zonally-averaged temperature and salinity differences between the isobarically
averaged WAGHC and WOA13 climatology. Using the isobarically averaged WAGHC, we tried to
minimize the effect of isopycnal averaging. Both temperature and salinity sections show the WAGHC
climatology being on average warmer and saltier. A rather pronounced dependence on latitude is observed,
with "tounges" of positive differences linked to the Antarctic Circumpolar Current and to latitudes north of
30$^\circ$ N.

The mean temperature difference for the layers 0 - 300m, 0 - 700m, and 0 - 1500m are 0.127, 0.079, and
0.048 $^\circ$C, respectively. We attribute these differences to real changes in the World Ocean over an
approximately 25 year time period between the WAGHC and WOA13 climatologiesy . The time difference



plot (Fig. 14c, Fig. 15c) was produced assuming the meadian year 1984 for the WOA13 climatology, which was created as the average of six decadal climatologies.

**9.4 Annual cycle**

Both the WAGHC and the WOA13 climatologies provide monthly temperature and salinity fields,

which were used to produce the annual cycle amplitude maps for temperature and salinity (Fig. 16). Both climatologies produce very similar amplitude patterns, with the highest temperature amplitudes found in middle latitudes of the Atlantic and  North Pacific oceans and in the tropical and equatorial belts. Maximum salinity amplitudes are observed in the Polar ocean and in several tropical areas like Indonesian seas and the
Northern Indian Ocean. The difference plots for temperature  (Fig. 16e) are characterized by  higher WAGHC amplitudes in the tropical belt, Gulf Stream, off the North-East Greenland, and within the Agulhas Return Current. The difference plot for salinity (Fig. 16d) shows generally much higher WAGHC amplitudes for the Polar Ocean and for the eastern tropical Pacific Ocean.

The zonally averaged September minus March differences shown in Fig.17 are very similar to the plots
based on Argo data and presented by Roemmich et al. (2009), and confirm the hemispheric asymmetry with seasonal amplitude in the northern hemisphere being much higher compared to the southern hemisphere. However, several systematic differences between the climatologies may be noted. The WAGHC climatology gives a 0.5ºC higher temperature amplitude  near the equator within the depth layer 50-100 meters. In comparison, the WAGHC climatology between 10 and 80ºN is characterized by a 0.2-0.5ºC lower amplitude
in the seasonal cycle. The annual cycle differences between the climatologies for salinity are less pronounced, with the largest differences found in the polar latitudes of the both hemispheres.

**9.5 Ocean Heat Content time series**

Finally, we used the WAGHC and WOA13 climatologies to test them as the base-line mean for the
calculations of the ocean heat content anomaly (OHCA) time series. Here the OHCA time series between 1920 and 2016 (Fig. 18) were calculated as follows. First, the depth averaged temperatures for the layers 0 - 300m and 0 - 700 m are obtained. The mean layer temperature anomaly is then differenced from a baseline climatological monthly mean. The global temperature anomaly for each layer is represented as the area-weighted mean of all 1-degree latitude zones containing data. For each 1-degree zone, the temperature
anomaly is represented by the mean of all 1-degree boxes containing data. The calculated global temperature anomalies are converted to OHCA over the entire ocean area. This is equivalent to the assumption that the mean temperature anomaly for the ocean boxes without data is equal to the mean anomaly estimated for the grid boxes with observations.  We note that the time series presented in Fig.18 represent the decadal mean anomalies centered on each calendar year. Fig 18 c-i shows temperature anomalies averaged for selected
decades in 1x1-degree boxes.

The irregular data sampling is the largest source of uncertainty in  global OHCA calculations. In order to estimate this kind of uncertainty, we used the global GECCO ocean synthesis (German contribution to Estimating the Circulation and Climate of the Ocean) (Köhl and Stammer, 2008). The method was applied earlier to the upper-ocean temperature anomaly calculations (Gouretski et al.,  2012). The GECCO synthesis
provides an estimate of  ocean circulation consistent with the dynamics of an ocean general circulation model.  The depth-averaged decadal temperature time series for the layers 0 - 300m and 0 - 700m were calculated from GECCO output (1) using boxes sampled in the historical record during each particular



decade and (2) using the full model output. The standard deviation of the difference between the two time series provides the measure of uncertainty due to the irregular and incomplete sampling for that decade.

Unfortunately, the historical climatologies are based on data irregularly distributed in time and space and can have different mean years for different regions of the ocean, thus introducing inconsistencies among the regions. Boyer et al (2016) used three monthly mean temperature climatologies to test the sensitivity of the OHCA estimates for the Global Ocean to the choice of the baseline mean. The OHCA uncertainty for the layer 0-700m due to the baseline mean was found to depend on the mapping method and time periods,

varying between 2.7 and 24.5ZJ, which corresponds approximately to 2 to 16 % of the full OHCA range between 1970 and 2010.

Our calculations revealed much smaller differences due to the choice of baseline mean.  The largest differences reach about 10% of the full OHCA range for some years between 1920 and 2015 and are observed before the mid-1950s. This time period is characterized by an extremely uneven distribution of

observations and almost no observations in the Southern hemisphere, especially during the 1940s. After the mid-1950s, the differences due to the baseline mean do not exceed a few percent.

We find an OHCA increase of ~150 ZJ since 1920 for the layer 0 - 300m and of ~220 ZJ for the layer 0 - 700m.  Both time series are characterized by an acceleration of the ocean heat content growth since the mid-1990s. As mentioned in section 9.3, the WAGHC and WOA13 climatologies have the mean year difference

exceeding 25 years, so that the overall temperature differences between the two climatologies can be attributed to climate change in the ocean over this time period. The overall temperature (OHCA) differences between the two climatologies are shown in green in Fig.18 and in both cases they are lower than the OHCA differences obtained from decadal time series (72% and 89% for the layers 0-300m and 0-700m respectively).

**10 Conclusions**

The paper introduces and describes in detail the new WOCE-Argo Global Hydrographic Climatology (WAGHC). The climatology was concieved as the update of the former WOCE Global Hydrographic Climatology, WGHC (Gouretski and Koltermann, 2004). Unlike its predecessor, the new climatology has a finer one-fourth degree spatial resolution and resolves the annual cycle of temperature and salinity on a

monthly basis. Two versions of the climatology are available, with the spatial interpolation being performed on  isobaric and isopycnal surfaces respectively.

The WAGHC climatology is further compared to the widely used one-fourth degree resolution isobarically averaged climatology WOA13, produced by the NOAA Ocean Climate Laboratory  (Locarnini et al, 2013). We note generally good agreement between these two gridded products. The differences between the two

climatologies are attributed basically to interpolation method (isopycnal versus isobaric averaging) and to the considerably improved data basis (the WAGHC includes additional four years of the Argo float  and other data). Inclusion of additional data into the WAGHC climatology significantly improved the representation of the thermohaline structure in polar regions.  However, a significant improvement was also achieved for several data abundant regions like the Baltic sea, Caspian sea, Gulf of California, Caribbean Sea, and the

Weddell Sea. Further investigations are needed to identify the causes of differences between the two climatologies in these regions.

We also tested the dependence of the ocean heat content anomaly (OHCA) time series on the baseline climatology. Since the 1950s, both WAGHC and WOA13 used as baseline means produce almost identical




OHCA time series. Even for the earlier data-poor decades, the largest differences do not exceed 10% of the
full OHCA range.

*Data availability.* The long-term data storage of the WAGHC climatology is provided by the Climate and
Environmental Retrieval and Archive (CERA) system hosted and maintained by the German Climate
Computing Center (DKRZ). The gridded climatology is available online at the Integrated Climate Data
Center-ICDC, which is part of the Center for Earth System Research and Sustainablity,

(http://icdc.cen.uni-hamburg.de/1/daten/ocean/waghc).

**Appendix A: Quality control tests on temperature and salinity profiles**

**1 Crude Range check**

The data are screened for extreme temperature and salinity values. Global temperature-depth and salinity-
depth histograms are used to define the respective masks for gross errors. Values falling outside the mask fail
the test. It is assumed that observations which failed the test give no information on the true parameter
values.

**2 Spike check**

The check aims to identify spikes on temperature and salinity profiles. For each triple of parameter values on
neighbouring depth levels $p_k$, $p_{k+1}$, $p_{k+2}$ the following test values are calculated:

s1=| $p_{k+1}$ -( $p_k$ + $p_{k+2}$ )*0.5|

s2=| $p_{k+2}$ - $p_k$ )*0.5|

s=s1-s2

If the value s exceeds the depth dependent threshold value $s_{max}$, the level k+1 is flagged. The test is not
performed for profiles with large gaps between the observed levels.

**3 Constant value check**

The test proves how many temperature/salinity measurements of one and the same profile are identical. The
test includes two tunable parameters: the minimal thickness of the layer within which all measurements
shows exactly the same parameter value, and the number of such levels within the layer. The first parameter
sets the threshold thickness of the thermo- and halostad, whereas the second parameter take into account the
typical observed level spacing, which differs between instrumentation types.

**4 Multiple extrema check**

This test identifies profiles with unrealistically large numbers of local parameter extrema. For each triple of
three neighbour observed levels the extremum is considered to be significant if  | $p_k$ - $p_{k+1}$ |<d and   | $p_k$ - $p_{k-1}$ |



<d, where the parameter d is selected to be larger than the measurement precision and the typical amplitude of the micro-scale parameter inversions.

## 5 Vertical gradient range check

This test identifies pairs of levels $k$ and $k+1$ for which the vertical gradients of temperature or salinity exceed the overall depth dependent ranges. The gradient ranges are defines on the basis of the depth-gradient histograms. Both observations are flagged when the vertical gradient falls outside the range.

## 6 Local climatological range check

For the calculation of the climatological parameter ranges the adjusted boxplot method for skewed

distributions is used (Vanderviere and Huber, 2004). Here, the skewness of the local parameter distribution is taken into account, so that the local climatological range is defined as

$[Q1 - H_l(MC) \, IQR; \, Q3 + H_r(MC)IQR,$

where $H_l(MC) = 1.5 \, e^{aMC}$, $H_r(MC) = 1.5 \, e^{bMC}$, $Q_1$ and $Q_3$ are the first and the third quartiles respectively, $IQR = Q_3 - Q_1$ is the interquartile range.

The medcouple MC is defined as:

$MC(F) = median \, h(x_1, x_2)$ , $(x_1 < MF < x_2)$, and $h(x_i, x_j) = [ \, (x_j - mF) - (mF - x_i) \, ] / ( \, x_j - x_i)$.

At each 0.25-degree grid node and at each standard level, the local median and the medcouple

were calculated using data within a variable influence radius. The influence radius was increased iteratively from the initial value of 55 km to the limit of 333 km in order to achieve the target number of 300

observations.

## 7 Sample depth vs local digital bathymetry check

For the local bathymetry check the 0.5-arcminute resolution digital GEBCO bathymetry was used. Profiles situated on land acording to the digital bathymetry were rejected. For the ocean profiles the levels deeper than the local bottom depth (added by the depth-dependent tolerance) were flagged and not used for the

further analysis.

## 8 Percentage of rejected (flagged) observed levels

Finally, all profiles with the percentage of flagged levels exceeding 80 % were rejected.


*Competing interests.* The author declares that he has no conflict of interest.

*Acknowledgements.* The ongoing efforts of the NOAA Ocean Climate Laboratory to prepare and disseminate the newest updates of the global hydrographic data archive which provided the main data basis for this study

are highly appreciated. I am thankful to the Department of Fisheries and Oceans of Canada for the



hydrographic data from the Arctic Ocean and the North Atlantic Ocean collected by the Freshwater Institute, Bedford Institute of Oceanography, Institute Maurice-Lamontagne, Northwest Atlantic Fisheries Centre, and Institute of Ocean Sciences. Particularly, I would like to thank Mathieu Ouellet,  Colline Combault, and Ives Gratton for preparing these data for me. My thanks go to the colleagues from the Alfred Wegener Institute,

Bremerhaven, and personally to Axel Behrend for sharing their collection of the Arctic hydrographic data with me. I am grateful to Marc Carson for careful reading of the manuscript and  numerous  improvement suggestions. Finally, I would like to thank the colleagues from the German Climate Computing Center (DKRZ) for their help in publication the WAGHC climatological gridded dataset. The work was conducted as part of the Excellence Initiative CLISAP at the Universität Hamburg, funded through the German Science

Foundation (Grant EXC 177/2).

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



**Figure captions**

Figure 1. Yearly number of profiles for each data type.

Figure 2. Temperature (a, b, c, d) and salinity (d, e, f, g) data rejection rates (all instrument types).

Figure 3. Example of the profile extrapolation procedure for three arbitrarily selected CTD temperature (a) and salinity (b) profiles.

Figure 4.  Mean absolute difference between the observed and extrapolated profiles for temperature (a) and salinity (b) at different merging depths.

Figure 5.  a) positions of full-depth profiles used for the extrapolation procedure (blue-before 1985, red -after 1984); b) positions of extrapolated profiles (red - Argo profiles, blue – non-Argo profiles); c) percentage of extrapolated levels; d) extrapolated profile frequency distribution vs the number of extrapolated levels; e) extrapolated profile frequency distribution vs the year of observation.

Figure 6. Area-mean climatological year versus depth. Monthly values above 1900 m are shown in red.

Figure 7. The mean average distance to four nearest bin-averaged profiles versus depth

Figure 8. Temperature (a, b, c) and salinity (d, e, f)  differences between the isopycnally-averaged and isobarically averaged WAGHC climatologies for selected depth levels in January: 150 m(a,d), 518m (b, e), 1050m (c, f); area-averaged differences vs depth (g).

Figure 9. Temperature (a, b, c) and salinity (d, e, f)   isopycnal WAGHC climatology  minus WOA13 climatology differences for selected depth levels in January: 150 m (a, d), 518m (b, e), 1050m (c, f)

Figure 10. Temperature (a) and salinity (b)  differences between the isopycnally-averaged and isobarically-averaged WAGHC climatology in the Northwestern Atlantic at 150 m level; (c,d): same but for the isopycnally-averaged WAGHC minus WOA13 differences.

Figure 11. T, S-histograms for six depth layers of the World Ocean. Bin-sizes are 0.05 ºC for temperature and 0.005 for salinity. Histograms are based on the gridded WAGHC and WOA13 climatologies. Only bins supported with at least 4 data points are plotted. Bins supported only by the WAGHC or WOA13 are shown in blue and red respectively. Overlapping bins are shown in green.

Figure 12. Selected areas within the World Ocean for which temperature-salinity histograms have been compared between the WAGHC and WOA13 climatologies.

Figure 13a. Temperature-salinity histograms for selected areas of the World Ocean (see Fig. 12) for WAGHC and WOA13 climatologies. Overlapping T, S-bins are shown in green;  T, S-bins populated only for the WAGHC climatology are shown in blue; T, S-bins populated only for the WOA13 climatology are shown in red. Bin-size is 0.1ºC x 0.05PSU

Figure 13b. (continuation)

Figure 14. Zonally-averaged differences  between the WAGHC and WOA13 climatologies for temperature (a), salinity (b), and mean climatological year (c).

Figure 15. differences  between the WAGHC and WOA13 climatologies versus depth for temperature (a), salinity (b), and for the  mean climatological year (c).



Figure 16. Annual cycle amplitudes for temperature (a – WAGHC, b - WOA13)  and salinity (d – WAGHC, e – WOA13) averaged over the upper 100 m layer for the  WAGHC and WOA13 climatologies.  Amplitude difference WAGHC minus WOA13 for temperature (c) and salinity (f).

Figure 17. Zonally averaged September minus March differences versus depth  for: a) WAGHC temperature, b) WOA13 temperature ; c) difference a-b; d) WAGHC salinity e)WOA13 temperature ; f) difference d-e

Figure 18. Decadal globally integrated ocean heat content anomaly (ZJ) time series for 1920-2016  for 0-300 m (a) and 0-700 m (b) layers computed using WAGHC (red curve)  and WOA13 (blue curve) baseline climatologies. Error bars correspond to the errors due to the irregular and incomplete sampling; the green 610 line corresponds to the heat content change estimated by differencing the WAGHC and WOA13 climatologies; (c-j) 0-300 m layer temperature anomalies in 1x1-degree boxes averaged over the selected decades.


Table 1. Instrumentation types and data sources contributing to the WAGHC climatology.

| Instrumentation type | Number of profiles | % all |
|---|---|---|
| Ocean Station Data profiles (OSD) | 2098823 | 44.452 |
| Conductivity-Temperature Depth profiles (CTD) | 971222 | 20.570 |
| Profiling floats (PFL) | 1368880 | 28.992 |
| Autonomous Pinniped Bathythermograph profiles (APB) | 282593 | 5.985 |
|  |  |  |
| Data source | Number of profiles | % all |
| World Ocean Database 2013 (WOD13) | 4665330 | 98.810 |
| Alfred Wegener Institute, Bremerhaven, Germany | 50848 | 1.077 |
| Canadian Institutions | 5340 | 0.113 |

Table 2.  Data rejection rate for the automated quality control procedure

| | | Data Type | | | | | | | |
|---|---|---|---|---|---|---|---|---|---|
| | | OSD | | CTD | | PFL | | APB | |
| | | Percent rejected levels | | | | | | | |
| Nr | Quality check | T | S | T | S | T | S | T | S |



| | | | | | | | | |
|---|---|---|---|---|---|---|---|---|
| 1 | crude parameter range check | 0.078 | 1.740 | 0.059 | 0.886 | 0.030 | 0.320 | 3.118 | 0.640 |
| 2 | spike check | 0.010 | 0.003 | 0.001 | 0.001 | 0.005 | 0.008 | 0.004 | 0.003 |
| 3 | constant value check | 0.004 | 0.114 | 0.007 | 0.114 | 0.004 | 0.128 | 0.024 | 0.043 |
| 4 | multiple extrema check | 0.092 | 0.009 | 0.056 | 0.043 | 0.062 | 0.074 | 0.129 | 0.046 |
| 5 | vertical gradient range check | 0.050 | 0.213 | 0.069 | 0.147 | 0.023 | 0.044 | 0.042 | 0.093 |
| 6 | local climatological range check | 2.669 | 5.533 | 2.375 | 3.180 | 1.079 | 1.746 | 7,088 | 9.170 |
| 7 | sample depth vs local digital bathymetry check | 2.877 | 2.877 | 3.588 | 3.588 | 0.165 | 0.163 | 9.268 | 9.268 |
| 8 | Percentage Levels Flagged | 5.517 | 0.793 | 2.048 | 2.645 | 0.197 | 0.680 | 8.421 | 7.204 |
| | Percentage rejected levels | 5.52 | 9.55 | 5.88 | 7.61 | 1.30 | 2.06 | 16.98 | 17.78 |
| | Percentage profiles with at least one rejected level | 20.52 | 29.99 | 23.98 | 27.78 | 14.09 | 15.93 | 35.69 | 50.13 |






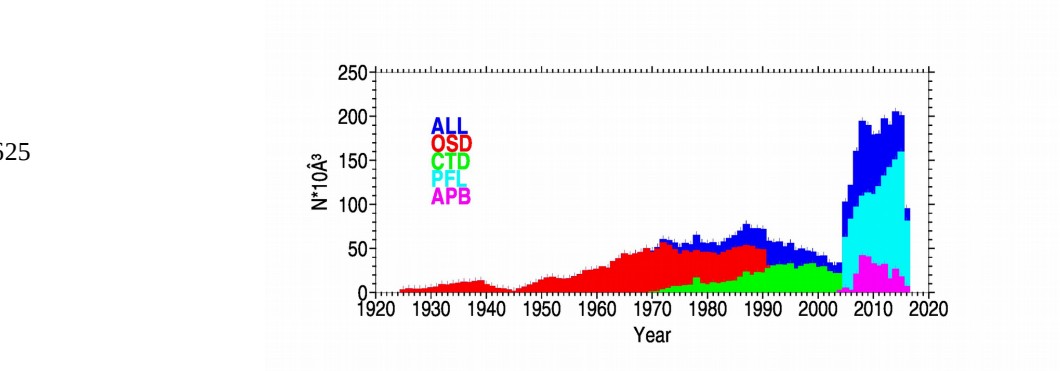

Figure 1. Yearly number of profiles for each data type.

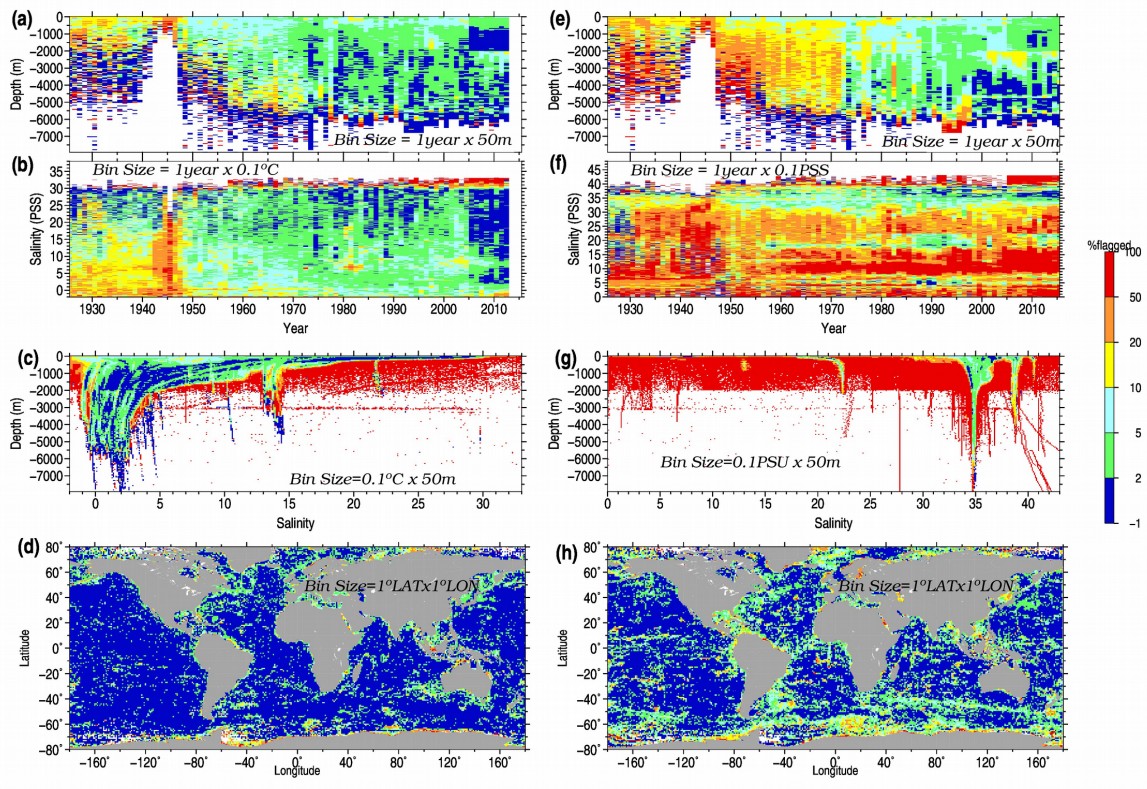

Figure 2. Temperature (a, b, c, d) and salinity (d, e, f, g) data rejection rates (all instrument types).




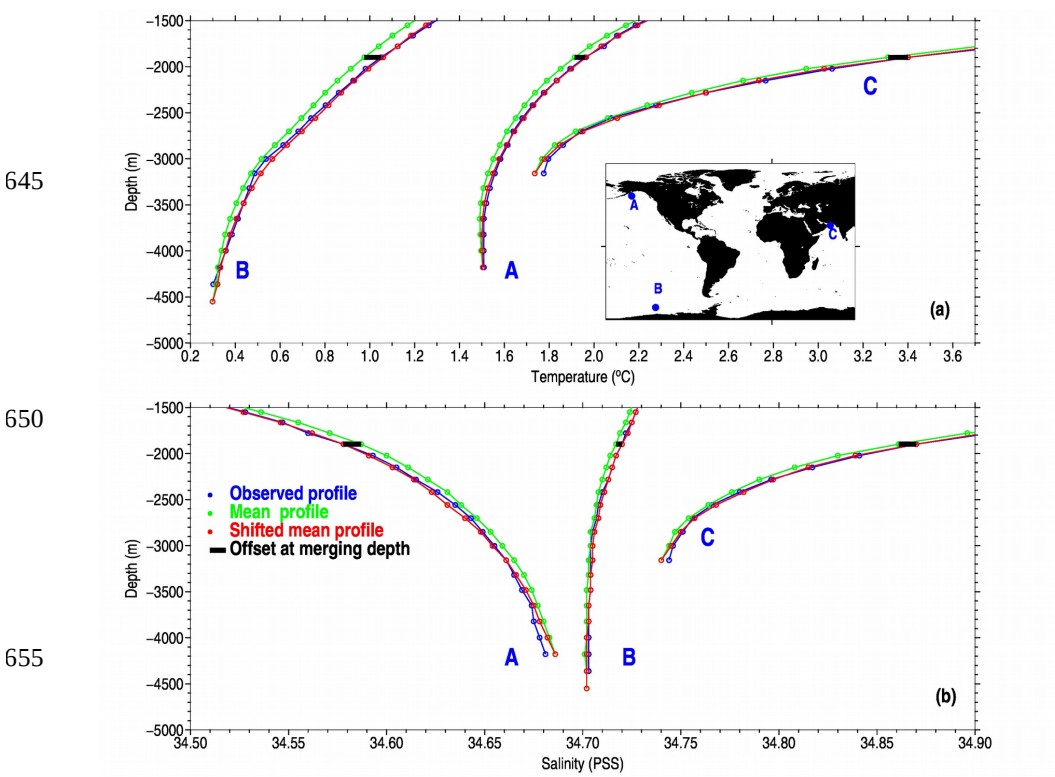



Figure 3. Example of the profile extrapolation procedure for three arbitrarily selected CTD temperature (a)
and salinity (b) profiles.





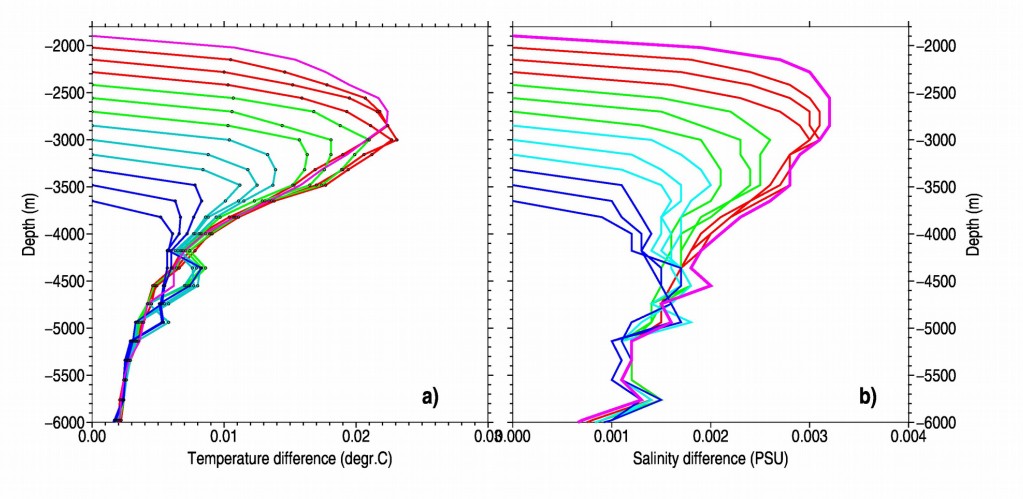

Figure 4. Mean absolute difference between the observed and extrapolated profiles for temperature (a) and salinity (b) at different merging depths



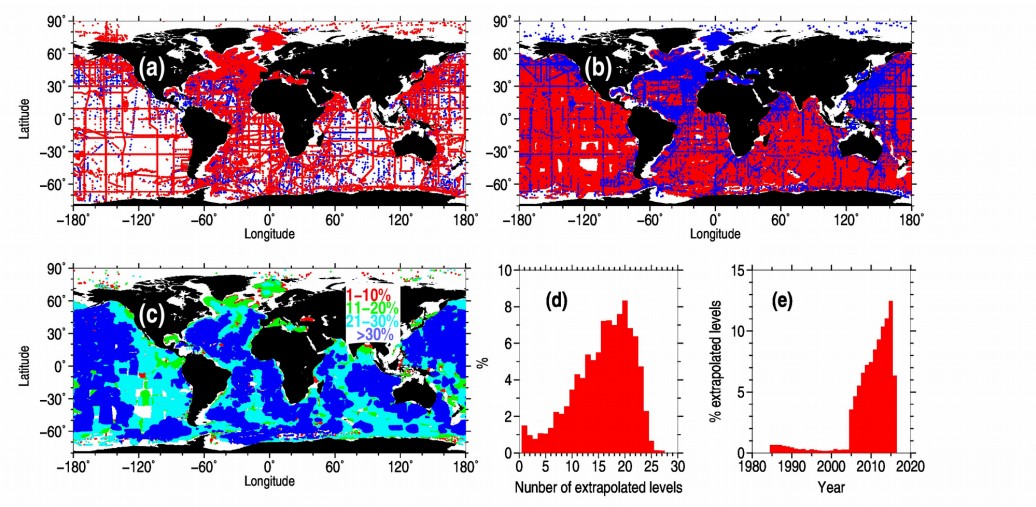

Figure 5. a) positions of full-depth profiles used for the extrapolation procedure (blue-before 1985, red -after 1984); b) positions of extrapolated profiles (red - Argo profiles, blue – non-Argo profiles); c) percentage of extrapolated levels; d) extrapolated profile frequency distribution vs the number of extrapolated levels; e) extrapolated profile frequency distribution vs the year of observation.


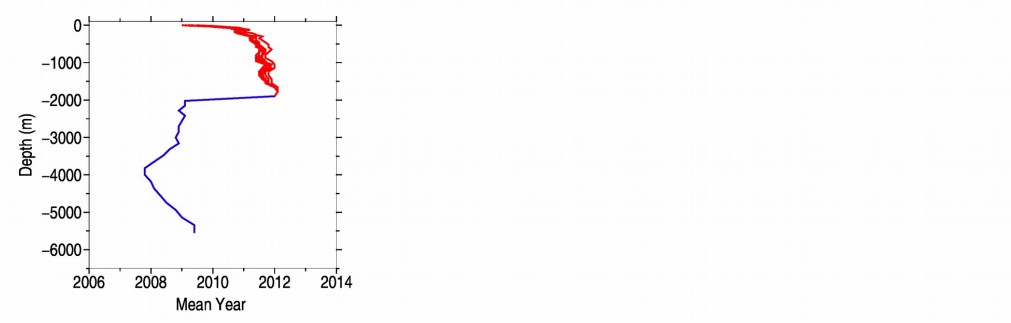

Figure 6. Area-mean climatological year versus depth. Monthly values above 1900 m are shown in red.






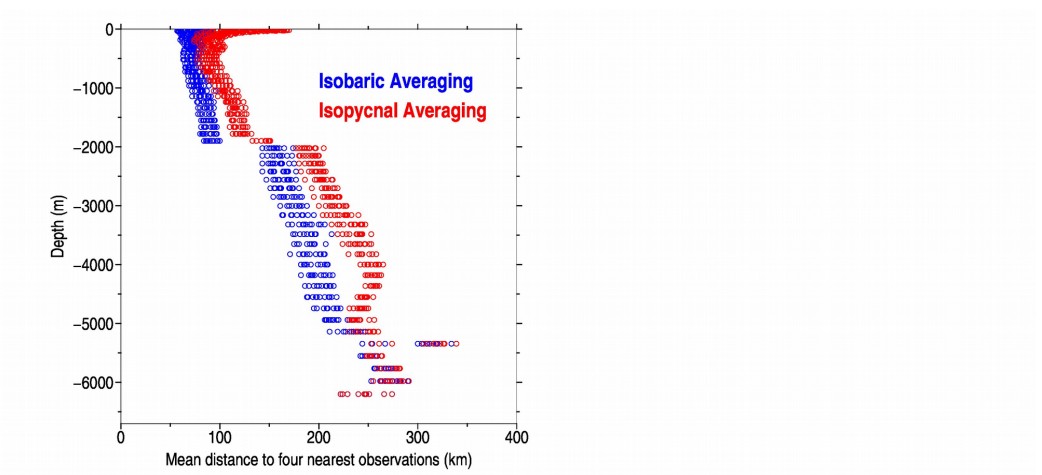

Figure 7. The mean average distance to four nearest bin-averaged profiles versus depth.

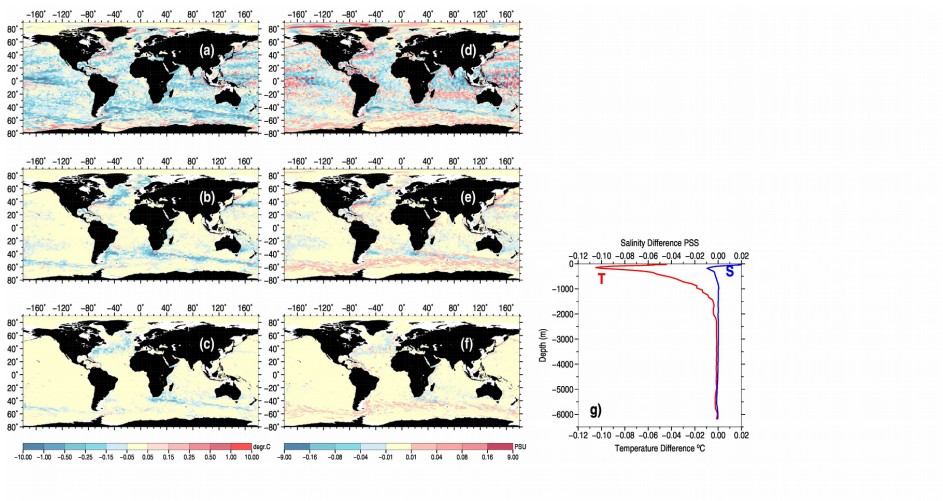

Figure 8. Temperature (a, b, c) and salinity (d, e, f) differences between the isopycnally-averaged and isobarically averaged WAGHC climatologies for selected depth levels in January: 150 m(a,d), 518m (b, e), 1050m (c, f); area-averaged differences vs depth (g).






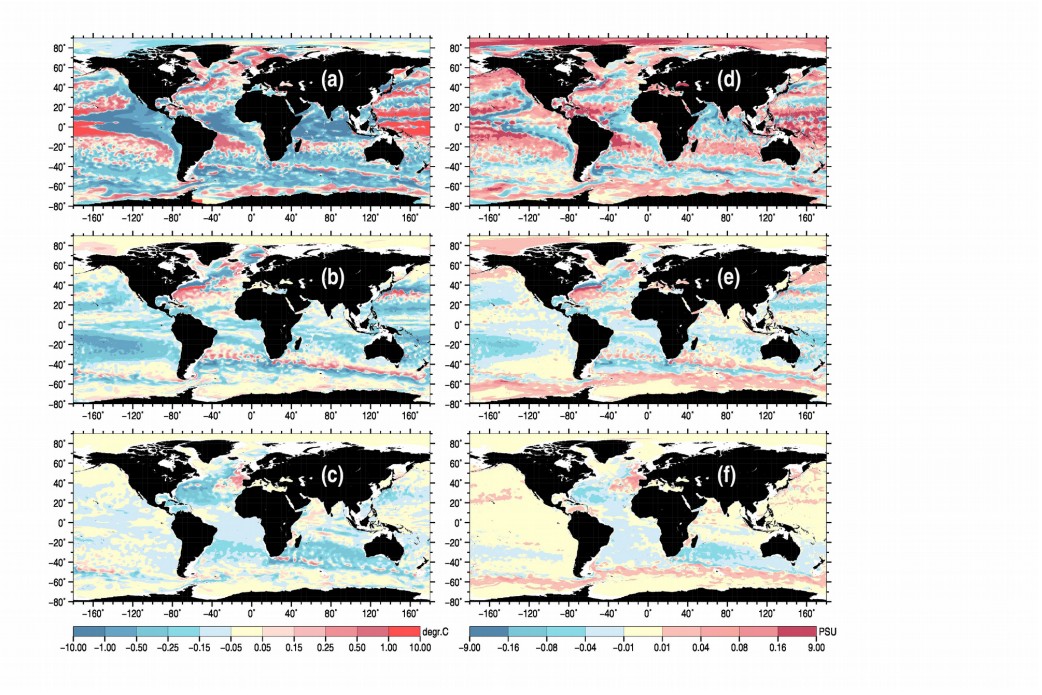

Figure 9. Temperature (a, b, c) and salinity (d, e, f) isopycnal WAGHC climatology minus WOA13 climatology differences for selected depth levels in January: 150 m (a, d), 518m (b, e), 1050m (c, f)

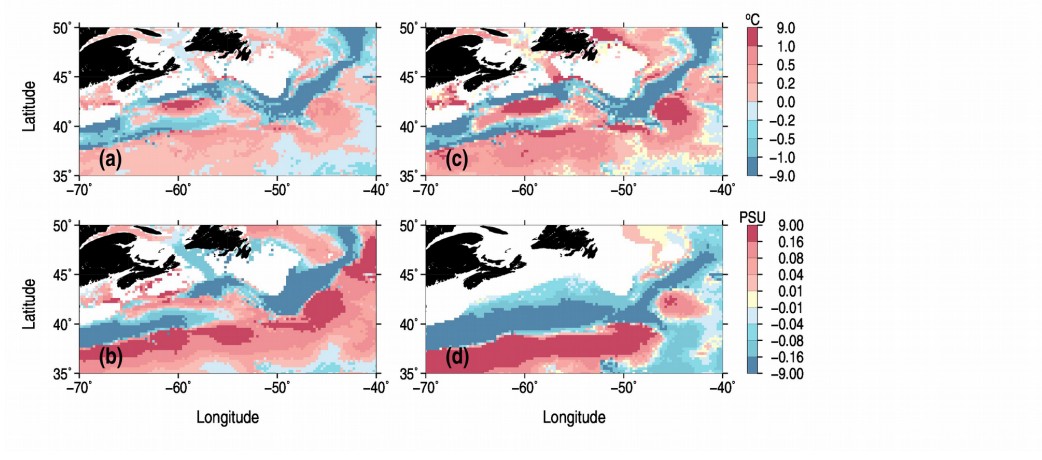

Figure 10. Temperature (a) and salinity (b) differences between the isopycnally-averaged and isobarically-averaged WAGHC climatology in the Northwestern Atlantic at 150 m level; (c,d): same but for the isopycnally-averaged WAGHC minus WOA13 differences.




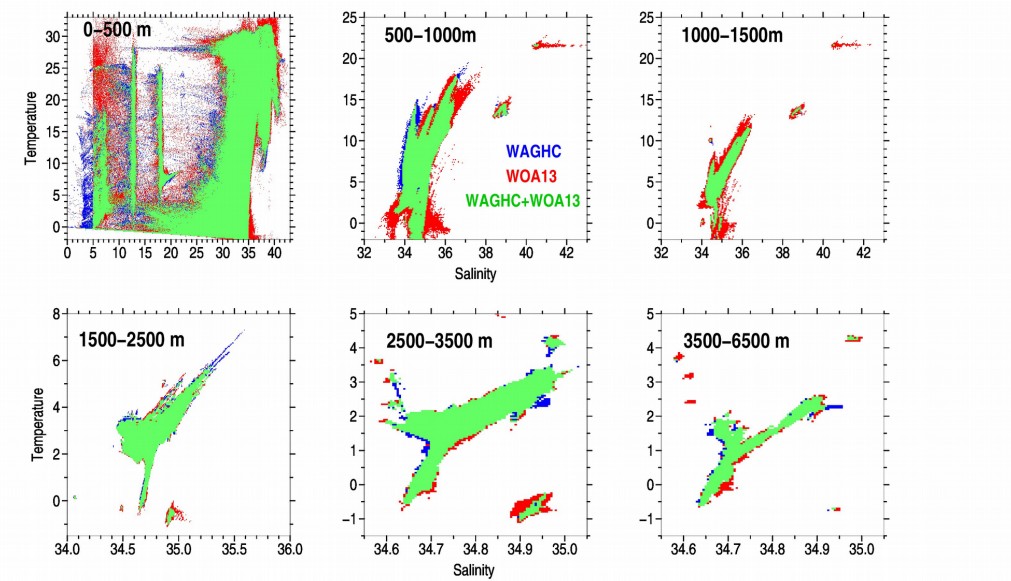

Figure 11. T, S-histograms for six depth layers of the World Ocean. Bin-sizes are 0.05 °C for temperature and 0.005 for salinity. Histograms are based on the gridded WAGHC and WOA13 climatologies. Only bins supported with at least 4 data points are plotted. Bins supported only by the WAGHC or WOA13 are shown in blue and red respectively. Overlapping bins are shown in green.






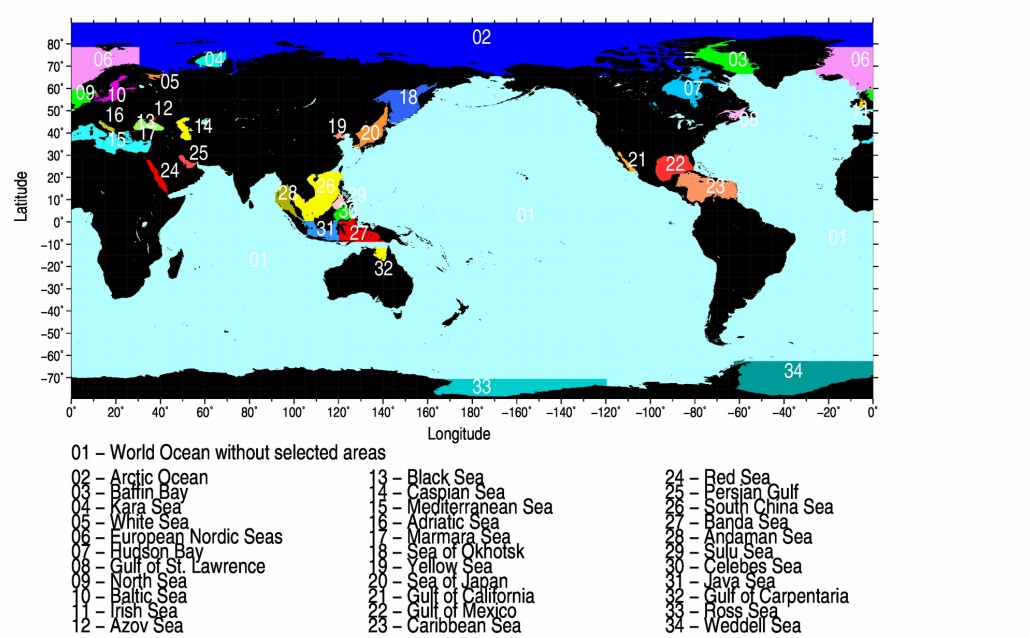

Figure 12. Selected areas within the World Ocean for which temperature-salinity histograms have been
compared between the WAGHC and WOA13 climatologies.



Figure 13a. Temperature-salinity histograms for selected areas of the World Ocean (see Fig. 12) for
WAGHC and WOA13 climatologies. Overlapping T, S-bins are shown in green; T, S-bins populated only
for the WAGHC climatology are shown in blue; T, S-bins populated only for the WOA13 climatology are
shown in red. Bin-size is 0.1oC x 0.05PSU




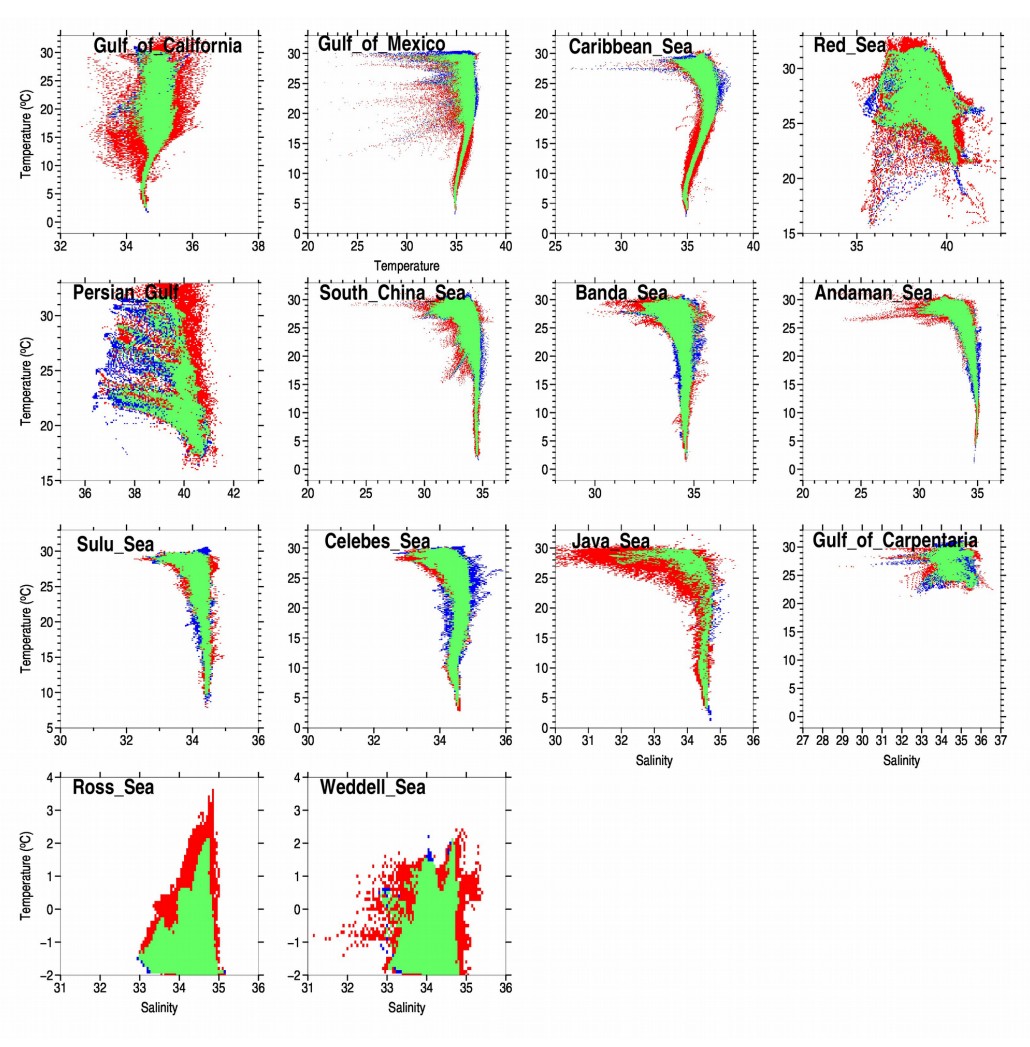

Figure 13b (continuation of the Fig. 13 a)



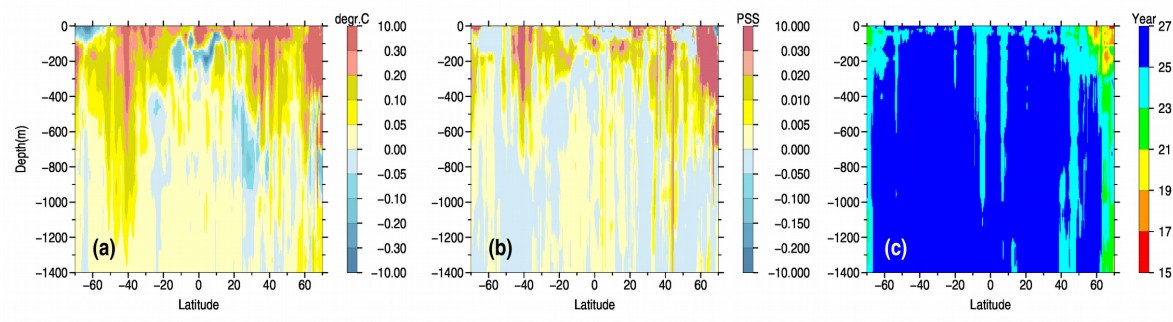

Figure 14. Zonally-averaged differences  between the WAGHC and WOA13 climatologies for temperature
(a), salinity (b), and mean climatological year (c).

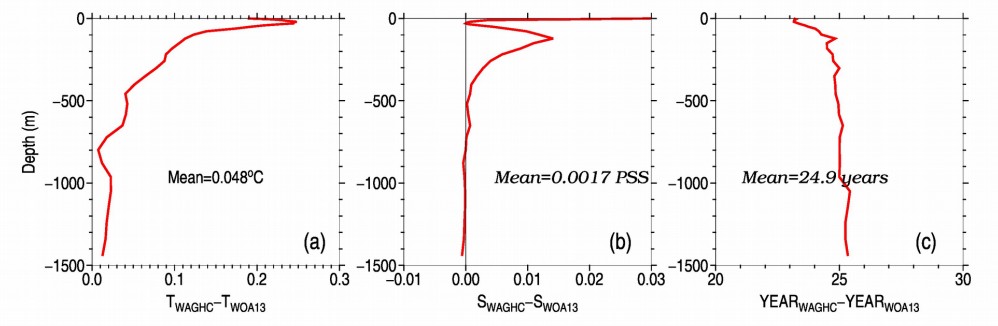

Figure 15. differences  between the WAGHC and WOA13 climatologies versus depth for temperature (a),
salinity (b), and for the  mean climatological year (c).



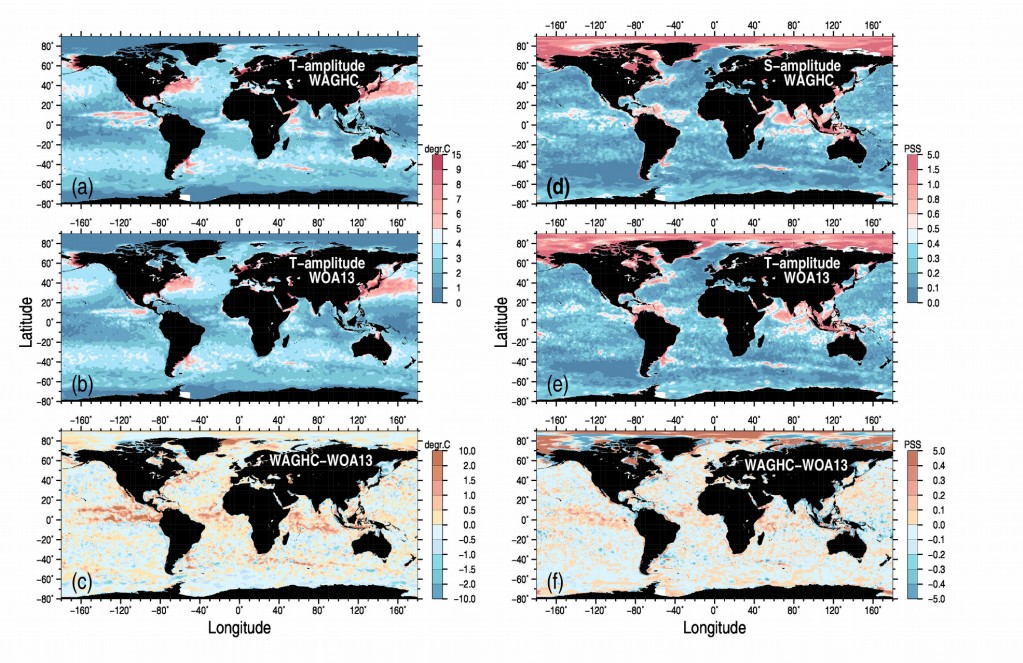

Figure 16. Annual cycle amplitudes for temperature (a – WAGHC, b - WOA13)  and salinity (d – WAGHC,
e – WOA13) averaged over the upper 100 m layer for the  WAGHC and WOA13 climatologies.  Amplitude
difference WAGHC minus WOA13 for temperature (c) and salinity (f).

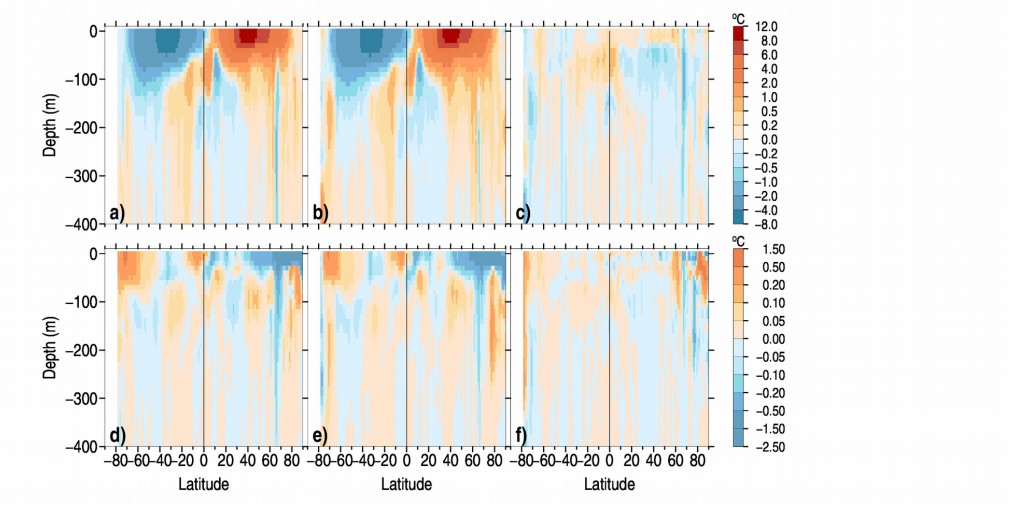




Figure 17. Zonally averaged September minus March differences versus depth  for: a) WAGHC temperature, b) WOA13 temperature ; c) difference a-b; d) WAGHC salinity e)WOA13 temperature ; f) difference d-e

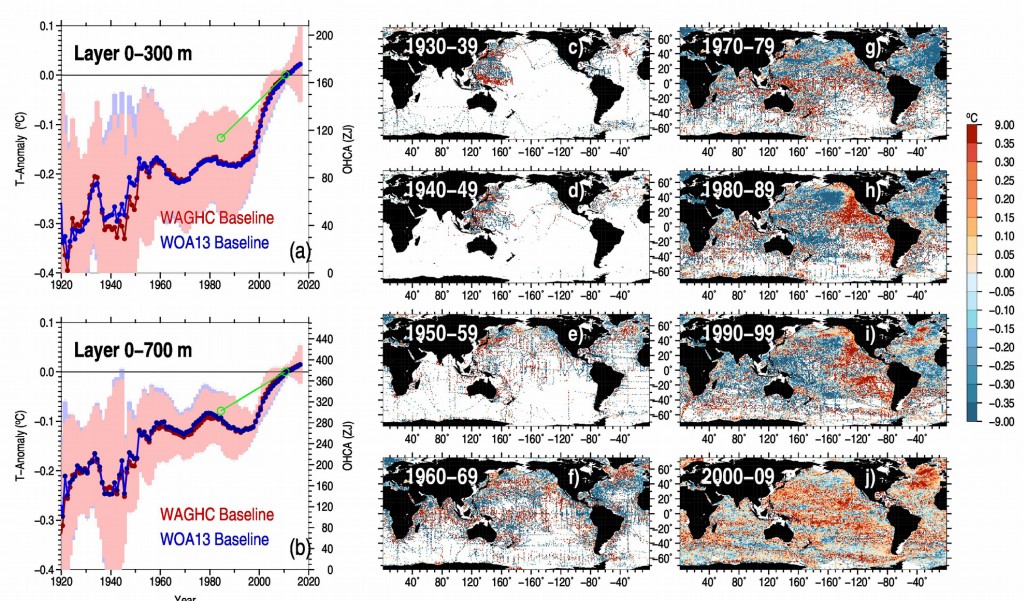

Figure 18. Decadal globally integrated ocean heat content anomaly (ZJ) time series for 1920-2016  for 0-300 m (a) and 0-700 m (b) layers computed using WAGHC (red curve)  and WOA13 (blue curve) baseline climatologies. Error bars correspond to the errors due to the irregular and incomplete sampling; the green line corresponds to the heat content change estimated by differencing the WAGHC and WOA13 climatologies; (c-j) 0-300 m layer temperature anomalies in 1x1-degree boxes averaged over the selected decades.
