# Peer review of "WOCE-Argo Global Hydrographic Climatology"

_Ocean Science, 2018_

## Referee Comment (RC1) · Anonymous Referee #1 · 3 May 2018

Review of the manuscript "WOCE-Argo Global Hydrographic Climatology" by Viktor Gouretski submitted for publication in Ocean Science

The paper describes, in great details, the new gridded WOCE-Argo Global Hydrographic Climatology (WAGHC) and compares this new digital oceanographic product to the NOAA World Ocean Atlas 2013 (WOA13). Importantly, the database for both atlases is the same NOAA/NCEI World Ocean Data Base (WOD) with some addition of the profiles from AWI, Germany. Both WAGHC and WOA13 are compiled with 0.25°x0.25° horizontal grid resolution, which by all means can be considered as high-resolution for the current state of global observational oceanography. However, WAGHC was computed at both on isobaric or isopycnal surfaces, while WOA13 is provided only on isobaric, which is effectively at fixed depths also called depth levels. The time span of WAGHC, between 2008 and 2012, is much shorter compared to the six decadal climatologies of WOA13 beginning in 1955-1964.

The methods of data interpolation of WAGHC and WOA13 are significantly different. The most important difference is in using the weighted values of vertically binned data in WAGHC compared to a more traditional way of objective analysis in WOA13. Moreover, WAGHC employs extrapolation of data below 2 km, while WOA13 uses observed data interpolated on the standard depths levels everywhere, including the deep ocean.

Having two considerably different gridded in situ ocean data products provides a unique opportunity to address the key scientific question of paramount importance—how trustworthy are the high-resolution gridded ocean climatologies? If the two products with so many intrinsic differences yield similar results, the answer is positive and thus the entire research perspective of mapping World Ocean looks good. On the contrary, if the differences are too large, the in situ mapping of oceans would become problematic and may be viewed as at least controversial and not yet ready to be implemented.

From that perspective, the results are truly spectacular. Although there are several regions where substantial differences were noticed, overall the two products compare very well indeed. The most striking results are in getting almost identical ocean heat anomalies despite substantially different baseline climatologies. Overall, the paper provides a very important comparison of two digital products with elaborated discussion of the differences and explanation of thereof.

I have no doubts that the paper should be published in Ocean Science with the minor revisions that would address some issues and after some clarifications that, in my view, may improve the presentation. There are some issues though that author may want to address in the revised version.

1. The first and quite important issue is that the title and the text imply that adding Argo data is a novel feature, which is true relative to WGHC, but not to WOA13 that includes over a million of Argo profiles.

2. NODC was replaced by NOAA National Centers for Environmental Information (NCEI) and should be now referred as that (although it may be advised to mention

that NCEI is formerly NODC when it is first cited).

3. Line 75: The annual and seasonal filed in WOA13 extend to 5500m, not to only 1500m; monthly climatologies do extend only to 1500m.

4. WOD13 contains approximately 12.8 million temperature profiles and 5.4 million salinity profiles with a great deal of the profiles dated after 1957 (the earliest date in WAGHC). Why were only 4,665 profiles retained for WAGHC?

5. For QC, an automated procedure was developed. Although it sounds like this procedure covers most of the problematic spots of removing the outliers, it is well known that no automated QC can be sufficient 100%. WOA13 implements manual QC, which is an integral and important part of the overall QC in that Atlas.

6. Lines 175-178: It looks as WAGHC uses a different approach to averaging the data to generate a climatology if compared to WOA13. In Woa13, the data belonging to each decade are averaged within that particular decade, and the averaged climatology between 1955 and 2012 is computed as averaged over the six decades. Such method prevents biases toward more recent decades with much more data. Including 1957-1984 data in the averages skews the WAGHC climatology toward more recent years even further. The author should mention this important difference between WAGHC and WOA13 because it may contribute to the found differences between these two products.

7. Lines 249-252: In several regions, including the Gulf Stream, GINS, etc., NCEI has recently generated so-called regional climatologies (RC) that are far more accurate and in some cases have a higher resolution (0.1x0.1 degree) https://www.nodc.noaa.gov/OC5/regional_climate/ . Comparison to those RCs may reduce the discrepancies. Moreover, similarly to numerical models, the differences between isobaric and isopycnal mapping decrease with increase in resolution. Although it is not realistic for the author to compare his results in selected regions with existed RCs in the reviewed research, it would be good for the OS readership to learn that

such option does exist and the results may be somewhat different, although probably not dramatically.

8. With regard to NCEI RC and ocean heat content, it worth to mention that finer-resolution mapping leads to significant changes in OHC spatial distribution compared to the one-degree resolution (Seidov et al, GRL, 44, p. 4985-4993, 2017). Moreover, the author did not elaborate on why 0.25x0.25 grid is critical for improvement ocean climatologies, as discussed in (Boyer et al, Int J of Climatology, 25(7), 931-945, 2005).

9. Lines 261-262: This is a false statement. WOA13 on 0.25x0.25 grid is available at all 102 levels, they are NOT replaced by 1x1 degree below 1500m; it is true, however, that monthly fields are limited to the upper 1500m only.

10. Although the overall quality of writing is outstanding, English can still be slightly improved. One common error is the absence of commas after introductory clauses. For example, "Since the late 19th century, the . . ." (line 27) requires a comma after "century"; there are a number of similar mistakes. Although all these comments are not critical, proper addressing them may further improve the manuscript.

To summarize, I recommend publishing this manuscript in Ocean Science after minor revisions.

---

## Referee Comment (RC2) · Anonymous Referee #2 · 10 Jul 2018

This manuscript describes the new gridded WOCE-Argo Global Hydrographic Climatology (WAGHC). Details of the method are provided along with a comparison between WAGHC and the NOAA World Ocean Atlas 2013 (WOA13). While both products are mapped on isobars with a 1/4 degree resolution, WAGHC is also mapped on isopycnals for comparison. This comparison is particularly valuable to understand the effect of averaging oceanic properties on isopycnal surfaces versus isobars (which may result in the production of water masses with temperature-salinity characteristics different from those of the observed data due to the non-linearity of the equation of state for seawater).

This work is a great contribution and the writing is outstanding. I recommend to accept the manuscript with minor revisions. I suggest to include these clarifications/modifications in the revised version of the manuscript:

- Line 135: It would be good to include (in the appendix) a plot of the depth dependent threshold value. Could the weighted-parabola method for vertical interpolation create unrealistic water masses for profiles with low vertical resolution ?

- Line 148: What threshold value is used ? How does it vary in space ?

- Line 254-255: How much of the difference (in Fig. 8) may be due to the estimate on isopycnals being biased towards the ocean interior where isopycnals outcrop ?

- Section 9.2 (Fig. 11, 13a-b): In place of Fig. 11, 13a-b, I suggest to show figures of the difference in volume (in each T/S bin) between WAGHC and WOA13, divided by the total volume in the same bin in WAGHC. This would provide more information on the differences in water masses between the two products.

Typos:

- Line 434: missing parenthesis
- Line 451: "defines" instead of "defined"
- Fig. 5d: "Nunber" instead of "Number" in xlabel
- Fig. 12: I suggest to include what different numbers mean in the caption.

---

## Author Comment (AC1) · 20 Aug 2018

I am very thankful for the detailed comments on the submitted manuscript. The following changes have been done:

1. It is explicitly noted in the text now (end of section 1) that the inclusion of teh Argo data is NOT the novel feature of the WAGHC climatology. The title and the name of the gridded data set simply stress the importance of the Argo data.

2. I have changed the NODC to NCEI in the Introduction.

3. The sentence was changed.

4. The difference between the total number of profiles in teh WOD13 and the number of profiles retained in the WAGHC is explained now.

[Figure]

5. I have added the following sentence: "The implementation of the manual quality control was restricted only to several areas in the Arctic Ocean with very poor data coverage".

6. I note in section 9.3 that the WOA13 has 1984 as the median year. This is different to the variable (with depth) median year for the WAGHC climatology (see fig. 6)

7. I have added a piece of text in the end of Section 9.1 with the reference to the respective 1/10-degree resolution climatology produced by NCEI for the Northwestern Atlantic. The comparison with that climatology could be the topic of another work.

8. I have added a piece of text regarding the choice of the spatial resolution in the end of the section 2. I have also included the reference to the work by Boyer et al (2005) where a more detailed elaboration on the issue can be found.

9. Thank you for pointing to this obvious error. We have changed the text respectively.

10. I have tried to do my best in proving the text again.

---

## Author Comment (AC2) · 20 Aug 2018

Dear referee, thank you very much for your comments on teh submitted manuscript. Below are my detailed answers to your comments:

Line 135: I have not included the plot of the depth dependent threshold ratio, as the depth-dependency is represented by a simple linear function. This is indicated in the text now.

Line 148: I have provided values for the threshold values for T and S.

Section 9.2. According to your suggestion I have replaced Figs. 11, 13a-b, which now show the volume fraction of the both climatologies for each T,S bin.

Line 434 - corrected Line 451 - corrected Fig 5d - corrected Fig 12 - the information is available in the figure caption now.